# Physician Detection of Clinical Harm in Machine Translation: Quality Estimation Aids in Reliance and Backtranslation Identifies Critical Errors

**Nikita Mehandru**[1], **Sweta Agrawal**[2], **Yimin Xiao**[2],
**Elaine C Khoong**[3], **Ge Gao**[2], **Marine Carpuat**[2], **Niloufar Salehi**[1]
[1]University of California, Berkeley  [2]University of Maryland
[3]University of California, San Francisco
nmehandru@berkeley.edu, sweagraw@umd.edu, yxiao@umd.edu
elaine.khoong@ucsf.edu, gegao@umd.edu
marine@umd.edu, nsalehi@berkeley.edu

## Abstract

A major challenge in the practical use of Machine Translation (MT) is that users lack guidance to make informed decisions about when to rely on outputs. Progress in quality estimation research provides techniques to automatically assess MT quality, but these techniques have primarily been evaluated in vitro by comparison against human judgments outside of a specific context of use. This paper evaluates quality estimation feedback *in vivo* with a human study simulating decision-making in high-stakes medical settings. Using Emergency Department discharge instructions, we study how interventions based on quality estimation versus backtranslation assist physicians in deciding whether to show MT outputs to a patient. We find that quality estimation improves appropriate reliance on MT, but backtranslation helps physicians detect more clinically harmful errors that QE alone often misses.

## 1 Introduction

Empowering people to decide when and how to rely on NLP systems appropriately is a critical, albeit challenging, endeavor. Appropriate reliance is a moving target because it is difficult to operationalize, and depends on context and application domain. Research in this space has so far relied on evaluations that are abstracted from real world use cases (Doshi-Velez and Kim, 2017; Narayanan et al., 2018; Boyd-Graber et al., 2022). We build on this work, and study appropriate reliance close to the actual decision that users have to make on the ground: whether or not to rely on a model output.

We study this question in the context of physicians deciding when to rely on Machine Translation (MT) in an emergency room when communicating discharge instructions to a patient who does not speak their language (Mehandru et al., 2022). MT represents an example of an NLP system used by millions of people in daily life (Pitman, 2021), in-

cluding in high-stakes contexts such as hospitals and courtrooms (Vieira et al., 2021). MT errors in those settings can be particularly harmful. In our setting, incorrect discharge instructions could lead to a patient misunderstanding their diagnosis or taking medications incorrectly, with potentially life-threatening consequences.

Research shows that people tend to over-rely on systems (Buçinca et al., 2021), and that explainability techniques that aim to address this issue can instead increase blind trust in incorrect predictions (Bansal et al., 2021). In the case of MT, most user studies have focused on human translators (Stewart et al., 2020; Castilho et al., 2019; Green et al., 2013), who have the expertise to evaluate MT faithfulness and to correct outputs when needed. Deciding how to rely on MT is much more challenging for people who use it to communicate in a language that they do not know. Zouhar et al. (2021) find that providing quality feedback to people using MT in travel scenarios has mixed effects, and can make them feel more confident in their decisions without improving their actual task performance.

This work evaluates the impact of quality estimation (QE) feedback on physicians' reliance on MT, building on decades of MT research on automatically estimating the quality of MT without reference translations (Blatz et al., 2004; Quirk, 2004; Specia et al., 2018; Fonseca et al., 2019; Han et al., 2021). However, QE systems are primarily trained to score overall translation quality outside of a specific context of use. It is unclear whether people will know how to interpret QE scores given prior evidence that they struggle to use probability and confidence estimates in explanations (Miller, 2018; Vodrahalli et al., 2022). Additionally, even when interpreted correctly, it is not clear whether seeing a QE score will make users better at deciding *when* to rely on an MT output and when not to, e.g. due to clinically relevant errors. We compare QE with

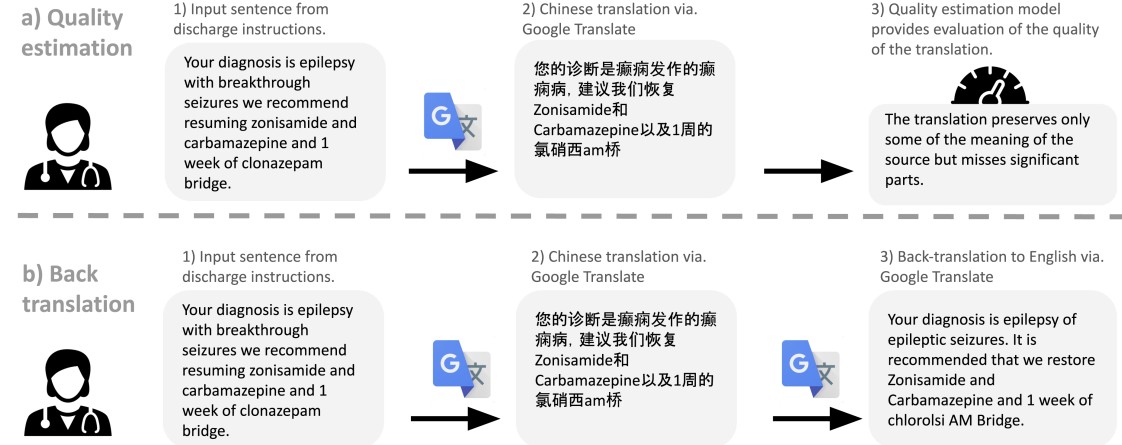

Figure 1: Physicians who participate in the study are shown a sentence from a real discharge instruction, and its translation to Chinese via Google Translate. They are asked to decide whether they would hand this translation to a patient who only speaks Chinese. Participants are randomly assigned to one of two conditions: a) Quality estimation: A quality estimation model provides an evaluation of the translation, or b) Backtranslation: The participant is shown the backtranslation of the text to English using Google Translate. We found that participants who were provided the quality estimation were better at detecting when to rely on a translation. For the most severe errors, backtranslation provided more reliable assistance. The sentence shown was marked as not adequate and life-threatening by bilingual physician annotators. The human reference translation for the Google Translate output states: Your diagnosis is seizure of seizure, suggest we resume Zonisamide and Carbamazepine and one week of Clonazepam-am-bridge.

a method commonly used by lay users, including physicians, to estimate the quality of a translation: backtranslation into the source language using the same system (Mehandru et al., 2022).

We conduct a randomized experiment with 65 English-speaking physicians to test how each of these interventions impacts their ability to decide when to rely on imperfect MT outputs (Figure 1). We find:

- The QE treatment group had a significantly higher confidence-weighted accuracy in their overall decision to give or not give a translation to a patient.
- The BT treatment group more effectively detected critical translation errors, those rated as having higher clinical risk.

In sum, both interventions improve physicians' ability to assess clinical risk and their confidence in their decisions, but for complementary reasons.[1]

## 2 Background

We situate this work in the MT literature before motivating our medical use case in Section 3.

**Clinical MT**  MT shared tasks motivated by medical applications, such as scientific abstracts or clin-

ical case translations, have led to research systems that produce translations that are more appropriate with in domain terminology than generic MT systems for diverse languages (Neves et al., 2022). However, in practical settings, clinicians turn to widely available MT systems such as Google Translate or Bing Translator (Randhawa et al., 2013; Khoong and Rodriguez, 2022). When translating Emergency Department discharge instructions from English into Chinese and Spanish, Google Translate was found to produce a majority of accurate outputs, however, a small number of inaccurate translations presented a risk of clinically significant harm (2% of sentences in Spanish and 8% of sentences in Chinese) (Khoong et al., 2019a). Our work evaluates tools that are used in practice by physicians and purposefully over-samples from error-prone sentences to present a useful evaluation framework that can also be used to evaluate dedicated clinical MT systems, complementing standard reference-based evaluation metrics which do not directly account for the potential clinical impact of MT errors (Dew et al., 2018).

**Quality Estimation**  Quality Estimation (QE), the task of automatically assessing the quality of MT outputs without access to human references, is a long-standing area of research in MT (Specia et al., 2018). State-of-the-art QE systems devel-

---

[1]Code and data to reproduce our findings are released at https://github.com/n-mehandru/PhysicianQE.git.

oped for evaluating MT such as OpenKiwi (Kepler et al., 2019), TransQuest (Ranasinghe et al., 2020), or COMET-src (Rei et al., 2020) are built on top of large scale multilingual models like BERT (Devlin et al., 2019) or XLM-RoBERTa (Conneau et al., 2020). They are trained to predict direct assessment scores collected by crowdsourced workers or post-editing efforts as measured by HTER (Snover et al., 2006). QE systems have been primarily evaluated "in vitro" by measuring the correlation of their scores with generic human judgments of quality collected independently from the context of use. QE has also proved successful at guiding human translation workflows (Stewart et al., 2020). However, it remains unclear how useful QE is for non-professional end-users in practice. In this work, we present assessments derived from the state-of-the-art COMET-src to physicians who do not speak Chinese to help them decide when to rely on English-Chinese MT.

**Backtranslation** We focus on scenarios where the input text is translated into a language unknown to the text's author. The onus to decide whether the output is acceptable therefore falls on the author, even though they do not have the expertise to assess translation quality. In these settings, people routinely use an intuitive feedback mechanism: backtranslation, which consists of using MT again to translate the original MT output in the input language, so they can compare it with the original in a language they understand. This practice has been decried in the MT literature (Somers, 2005) as backtranslation provides a very noisy signal by lumping together errors from the forward and backward MT pass, and potentially hiding errors that the backward pass recovers from. Nevertheless, people use backtranslation routinely, perhaps encouraged by interfaces that let them switch MT translation direction with a single click. However, little is known about the usefulness of backtranslation in these practical settings. Zouhar et al. (2021) conduct a user study evaluating the impact of backtranslation by lay users for the practical task of booking a hotel room in a language that they do not speak. They hypothesize that backtranslation feedback can help people craft the content of a message so it is correctly interpreted by the recipient. Backtranslation feedback was found to greatly increase user confidence in a translation, without improving the actual translation quality. We study the impact

of backtranslation in a clinical decision-making context and compare it to a QE model output.

## 3 MT for Cross-lingual Communication Between Patients and Physicians

Cross-lingual communication is imperative in the presence of language barriers between physicians and patients. We focus on a specific high-stakes context: helping physicians communicate discharge instructions to patients in the Emergency Department (ED). While medical interpreters often facilitate conversations between clinicians and Limited English Proficiency (LEP) patients, this has been found to be insufficient in helping patients recall what they are supposed to do after getting discharged (Hoek et al., 2020). Further, comprehension of emergency discharge instructions is known to be an important contributor to patient non-adherence (Clarke et al., 2005).

Our prior work has found that MT is frequently used in practice for tasks such as automatically translating discharge instructions (Mehandru et al., 2022), and thus provides a written record for patients to aid in comprehension at discharge time as well as recall and adherence. A key challenge is that it is difficult for physicians to ensure that patients comprehend written discharge instructions when they cannot verify the accuracy of a machine-generated translation. As an added complication, limited health literacy and discharge plan complexity can lead patients to overestimate comprehension (Glick et al., 2020).

Designing MT for effective physician-patient communication involves many stakeholders. This work focuses on physicians as a starting point, as findings can inform their training and strategies for cross-lingual communication to maximize impact in the short-term. We leave to future work the equally important question of helping diverse LEP patient populations rely on MT output adequately.

## 4 Methods

We conducted a randomized controlled experiment to test how quality estimation and backtranslation impact physicians' appropriate reliance on MT.

### 4.1 Emergency Department Discharge Instructions Data

**Source Text** English source text for our experiment is drawn from de-identified Emergency Department (ED) discharge instructions that were writ-

ten between the years 2016 to 2021 at the University of California, San Francisco (UCSF). We select six discharge instructions, and a total of twenty-eight sentences from these notes, to ensure that they present expected key elements in a discharge instruction to a patient, including presentation of the problem (chief complaint), actual diagnosis, medication list, follow-up items, and a 24/7 call-back number with the referring provider (DeSai et al., 2021; Halasyamani et al., 2006). Additionally, sentences were selected based on the source text complexity, and the presence of certain words having multiple meanings.

**MT** The subset of selected English sentences were automatically translated into simplified Chinese by Google Translate.[2] We chose English-Chinese, because it is a high-demand language pair that is often needed in clinical settings in the United States, and Google Translate is known to be used by physicians (Khoong et al., 2019b; Mehandru et al., 2022). While general translation quality is expected to be reasonably strong, on FLORES, the translation quality as measured by BLEU is 38.50 for the English-Chinese (Simplified) devtest (Goyal et al., 2022). Medical texts are typically out of the training domain, and as a result, clinically harmful translation errors have been documented for this specific language pair (Khoong et al., 2019b).

**Gold Annotation** Three bilingual physicians independently annotated each MT output along two dimensions: **translation adequacy** and **clinical risk**. Adequacy was defined as whether the Chinese translation accurately conveyed the meaning of the English source text (Turian et al., 2003). Physicians rated clinical risk based on the translation presented to them according to five categories: clinically insignificant, mildly clinically significant, moderately clinically significant, highly clinically significant, and life-threatening (Nápoles et al., 2015). During the annotations, physicians kept in mind how if a monolingual patient were to read the translation, whether or not the patient would understand the discharge instruction sentence. The three physicians then met with the lead author to discuss disagreements in the clinical risk ratings, and agree on a final label for each sentence.

---

[2]We use the Google Sheets Translation API.

## 4.2 Experimental Design

We conduct a between-subjects experiment with participants randomly assigned to one of the two treatment conditions: **backtranslation (BT)** and **quality estimation (QE)**. We added a **baseline** condition to assess participant responses to MT in the absence of feedback, which both groups completed first (within-subjects).

**Participants** We used convenience sampling to recruit sixty-five physicians to participate in our study. Medical residents were in training programs in the United States, and practicing physicians worked in multilingual settings. Their specialties included: internal medicine, cardiology, emergency medicine, neurology, surgery, family medicine, pediatrics, allergy and immunology, intensive care, obstetrics and gynecology, infectious diseases, military medicine, and psychiatry. 45% of physicians reported interacting with LEP patients daily, while another 45% responded interacting with them either two to three times per week or bi-weekly. 17% of physicians reported using Google Translate on a monthly basis when writing discharge instructions, while 14% responded using it bi-weekly or two to three times a week.

We randomized physicians into each condition for a total of thirty-five physicians in the backtranslation group and thirty in the QE group. Physicians were screened to ensure they were fluent in English, and had no knowledge of Chinese. This inclusion criteria for the experiment was in the demographic section of the survey, and was included in our recruitment emails. We further manually filtered out any participants who may have missed this statement and reported that they spoke Chinese in the pre-survey.

This study was approved by our Institutional Review Board (IRB), and physicians were compensated for taking the time to participate in our experiment.

**Survey Design** Physicians were first presented with the baseline condition. They were asked to read a discharge instruction, and then were presented with a sentence from the note and its respective Chinese MT translation. After the baseline condition, physicians were presented with one of the two treatment interventions, and the same set of twenty-eight English sentences from the six discharge instructions, each associated with its Chi-

| | ADEQUATE | | CLINICAL RISK | | | | |
|---|---|---|---|---|---|---|---|
| | YES | NO | INSIGNIFICANT | MILD | MODERATE | HIGH | LIFE-THREATENING |
| CONSISTENT | 10 | 4 | 8 | 4 | 2 | 0 | 0 |
| MOST PRESERVED | 4 | 6 | 3 | 2 | 3 | 2 | 0 |
| SOME PRESERVED | 1 | 3 | 0 | 2 | 1 | 0 | 1 |

Table 1: Each column represents the physicians' true labels on clinical risk (e.g., high clinical risk implies that the translation meaning was not similar to the original source). Each row represents the QE's assessment of the translation with consistent meaning that the translation preserved the original source's meaning. This chart shows QE quality labels are imperfect, but good enough to be potentially useful. The system errs by overestimating translation quality compared to human assessments of adequacy and clinical risk on our dataset.

nese MT translation except also accompanied by one of two quality feedback types.

In all conditions, after seeing a Chinese translation, participants were asked: 1) whether they would give the translation to a patient who only reads Chinese (binary question, yes or no), and 2) their confidence in this decision on a five-point Likert scale (1= Not Confident and 5= Very Confident). In the treatment conditions, participants were additionally asked to assess whether a monolingual Chinese patient would understand the discharge instruction sentence after reading the Chinese MT, using a five point Likert scale (1= Patient would understand none of the meaning and 5= Patient would understand all of the meaning).

Acknowledging the real-world context in which physicians would have to make these decisions, participants were asked to assume that a medical interpreter had already reviewed the discharge instruction with the Chinese-speaking patient. To ensure that they focused on patient comprehension of the discharge instruction as it pertains to clinical outcomes, physicians were also instructed to respond to questions without concern for lawsuits and regulatory requirements around showing an imperfect translation to a Chinese-speaking patient.

### 4.3 Treatment Conditions Details

**Quality Estimation** We adopt the state-of-the-art quality estimation (QE) system, Comet-QE (`wmt21-comet-qe-mqm`)[3] to design this treatment condition (Rei et al., 2021). Each source and MT pair is passed through the trained QE predictor to generate a score in the range of $[-1, 1]$. A positive score indicates that the translation quality of the sentence is better than average, while a negative score indicates below-average quality. While model-based predictions correlate well with human judgments (Freitag et al., 2022), they are hard to interpret. Hence, we partition the $[-1, 1]$ interval

to define quality labels, motivated by those used to assess translation quality in human evaluation of MT (Freitag et al., 2021; Kocmi et al., 2022).

To define these labels in a data-driven fashion, we collect a small development dataset by asking a bilingual physician to answer two questions about the quality of 125 sentences sampled from a related patient-physician conversational dataset:[4] a) *is the translation accurate?* b) *can the translation error pose a clinical harm?* (Fareez et al., 2022). We identify thresholds for QE scores on this development set based on the ROC curves for adequacy and risk prediction. The translation pairs are then labeled according to Table 2.

| RAW QE SCORE | OUR QE LABEL |
|---|---|
| $[0.101; 1]$ | The meaning of the translation is consistent with the source. |
| $[0.072, 0.101]$ | The translation retains most of the meaning of the source. |
| $[-1; 0.072]$ | The translation preserves only some of the meaning of the source but misses significant parts. |

Table 2: Thresholds for QE scores and their assigned labels.

Table 1 shows the breakdown of QE labels for clinical risk and adequacy assessment on the gold annotation data. COMET-QE achieves an accuracy of 66% and 73% on detecting adequate and clinically insignificant translations, respectively. However, COMET-QE was not effective at detecting incorrect translations that caused clinical risk. Of the nine translations that are deemed by bilingual experts to cause moderate to life-threatening harm to users (columns "moderate", "high", and "life-threatening"in Table 1), COMET-QE rated 2/9 as having "consistent" translations. The QE labels provided are thus accurate enough to be potentially

---

[3] https://github.com/Unbabel/COMET

[4] This dataset provides simulated patient-physician interactions in English across six medical cases, which are closer to discharge instructions than e.g., clinical notes aimed at other physicians rather than patients. The conversations are automatically transcribed and manually post-edited.

useful, yet realistically imperfect, as expected of automatically generated reliability measures.

Given the QE labels extracted using the above strategy, participants were then presented with a description of the quality estimation system, and the output labels they expect to see. They were asked to decide, based on the information provided, whether they would provide the translation to a patient and their confidence in the assessment.

**Backtranslation** In the backtranslation treatment condition, physicians were presented with the source English sentence, and were told that the Google Translation system generated a Chinese translation for the given sentence. Participants were then presented with text explaining that Google Translate translated the previous text back into English and were then shown the output translation.

## 4.4 Measures

Our goal in this work is to study whether physicians can more accurately rely on an MT output if they are provided with an evaluation of the quality by a QE model, or by seeing the backtranslation by the same MT system. Our outcome metric is as close to possible to the actual decision that physicians make in practice. We asked physicians to decide whether they would share a translation with a patient and also their confidence in that decision. To measure their overall performance, we use confidence weighting (Ebel, 1965), a common metric in cognitive psychology that measures whether the participant made the correct decision weighted by their confidence in that decision. Intuitively, confidence-weighted accuracy provides a way to encapsulate the properties of appropriate reliance in one metric: making accurate decisions and calibrating confidence in the model appropriately. In other words, if the participant makes an error with high confidence, this metric penalizes them more than if they make the same error but with less confidence.

**Reliance Metrics** Given P physicians, S instances (sentences), let $s^*$ be the correct answer for sentence s, and $\hat{s_p}$ is the answer selected by the p-th physician for this sentence s with confidence $c_s$ on a scale of 1 to 5. Our experiment uses the following measures:

The **Physician Accuracy (%)** for each condition (BT, QE) for each physician (p) is given by is:

$$Accuracy = \frac{1}{S} \sum_{s \in S} \mathbb{1}[s^* == \hat{s_p}] \qquad (1)$$

The **Confidence Weighted Accuracy (CWA)** for each condition (BT, QE) for each physician (p) is given by is:

$$CWA = \frac{1}{S} \sum_{s \in S} sign(s) \frac{c_s}{5}$$

$$sign(s) = \begin{cases} 1, & \text{if } s^* == \hat{s_p} \\ -1, & \text{otherwise} \end{cases} \qquad (2)$$

**Correctness** We define a correct decision by comparing the physician's decision with the adequacy of the translation deemed by our physician annotators.

*Adequate Translation.* An adequate translation is one in which the discharge instruction sentence was passed through Google Translate and annotated by bilingual English-Chinese physicians as correctly conveying the meaning of the English source text. An accurate decision in this context would be a physician giving the discharge instruction sentence to a monolingual patient.

*Inadequate Translation.* An inadequate translation is one in which the discharge instruction sentence was passed through Google Translate and annotated by bilingual English-Chinese physicians as incorrectly conveying the meaning of the English source text. An accurate decision in this context would be a physician not giving the discharge instruction sentence to a monolingual patient.

## 5 Results

We will show that physicians in the quality estimation (QE) condition had significantly higher confidence-weighted accuracy (CWA) than their baselines compared to those in the backtranslation (BT) condition. We also found that while the QE intervention increased their overall CWA, physicians in the QE treatment group were significantly worse than those in the BT treatment group at detecting errors, especially those labeled with higher clinical risk. We end with a discussion of the potential complementary roles that these two interventions can play for informed reliance on MT in high-stakes settings.

**QE Treatment Group has Higher Confidence-Weighted Accuracy** Physicians in the QE treatment group ($M = 19.6, SD = 13.7$) had a significantly higher CWA than the baseline for that group ($M = 11.2, SD = 15.3, t(58) = 2.1, p = 0.03$, Figure 2). We did not find any difference in

| DISCHARGE INSTRUCTION | CLINICAL RISK | ACCURACY (%) | |
| --- | --- | --- | --- |
| | | BT | QE |
| We believe this was due to something called vasovagal response which causes people to feel faint or lose consciousness. | moderately significant | 80 | 23 |
| Please try to drink fluids to make sure you are hydrated today. | moderately significant | 89 | 40 |
| Be sure to avoid taking NSAIDs (ibuprofen, naproxen) and Aspirin for pain. | mildly significant | 31 | 23 |
| The orthopedic surgeons drained your knee. | highly significant | 91 | 37 |
| You may take norco for pain, however, do not drink or drive as it will cause drowsiness. | moderately significant | 71 | 43 |
| Your diagnosis is epilepsy with breakthrough seizures we recommend resuming zonisamide and carbamazepine and 1 week of clonazepam bridge. | life-threatening | 97 | 80 |
| Please START taking and Doxycycline 100mg twice per day for seven days - Keflex 500mg four times per day for seven days | highly significant | 94 | 53 |

Table 3: A sample of the sentences used in our study, the clinical risk of their translation as determined by bi-lingual physicians, and participant accuracy rates across conditions for that sentence.

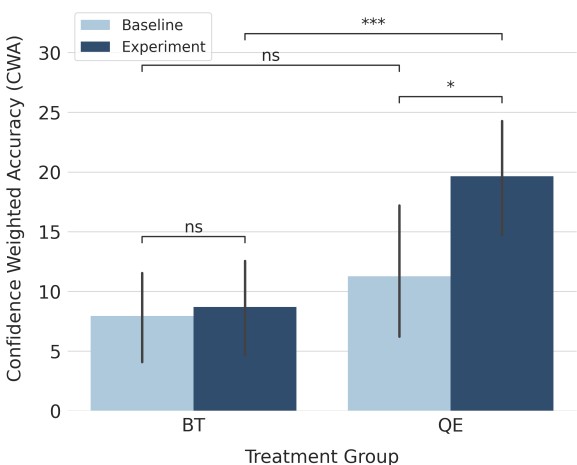

Figure 2: The QE treatment group had a significantly higher confidence-weighted accuracy score than the baseline condition for that group as well as the BT treatment group. However, we did not find any significant difference between the BT treatment group and the baseline condition for that group.
ns: not significant; ∗: significant with p-value $< 0.05$; ∗ ∗ ∗: p-value $< 0.001$.

CWA between the physicians in the BT treatment group ($M = 8.6, SD = 11.3$) and their baselines ($M = 7.9, SD = 11.4, t(68) = 0.2, p = 0.7$). This means that physicians significantly improved in their ability to rely appropriately on MT when presented with the QE evaluation, but not when presented with the BT. The difference between the QE and BT treatment groups was significant ($t(63) = 3.4, p < 0.001$).

For discharge instruction sentences that were labeled as inducing higher clinical risk (moderately clinically significant, highly clinically significant, or life-threatening), the BT treatment group identi-fied clinically harmful errors at a much higher rate. For example, consider the last sentence in Table 3, which gives medication instructions to a patient. The MT was annotated by bilingual physicians as *not adequate* and inducing *highly clinically significant risk*. Physicians in the BT treatment group correctly decided not to give the MT output to a Chinese patient with 94.3% accuracy, while the QE treatment group did so with only 53.5% accuracy.

**Post-Survey Analysis** In a post-survey, we asked physicians to rate their confidence in using the QE and Google Translation systems in a clinical workflow on a scale of 1 to 5. Physicians in the QE group reported higher levels of confidence in the quality estimation system ($M = 2.6$) compared to physicians in the BT ($M = 1.6$). A Mann-Whitney U test revealed a significant difference in responses between the groups, U = 229, p < .001. We saw a similar difference in the response to whether they would like to use each tool for clinical decision-making. Appendix Figure 5 has full post-survey results.

## 6 Discussion

We found that **QE and BT may play a complementary role**: QE can assist physicians in their decision to provide translated written discharge instructions to patients, while BT was more effective in detecting critical errors in MT systems. Combining aspects of BT and QE feedback may therefore benefit MT users in high-stakes settings. Our results show that, contrary to common MT wisdom, BT should not be entirely discounted as a quality feedback method for people, aligning with recent

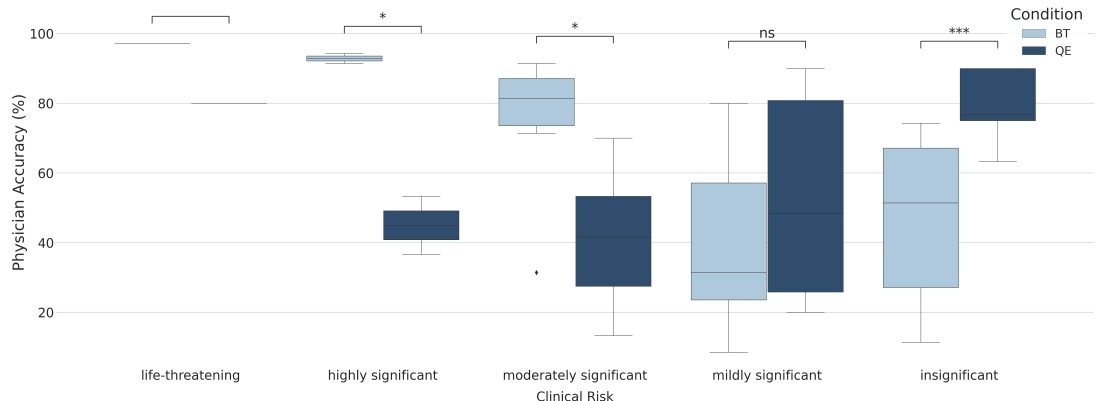

Figure 3: BT enables physicians to detect errors with high clinical risk at higher accuracy. ∗: p-value of 0.05;

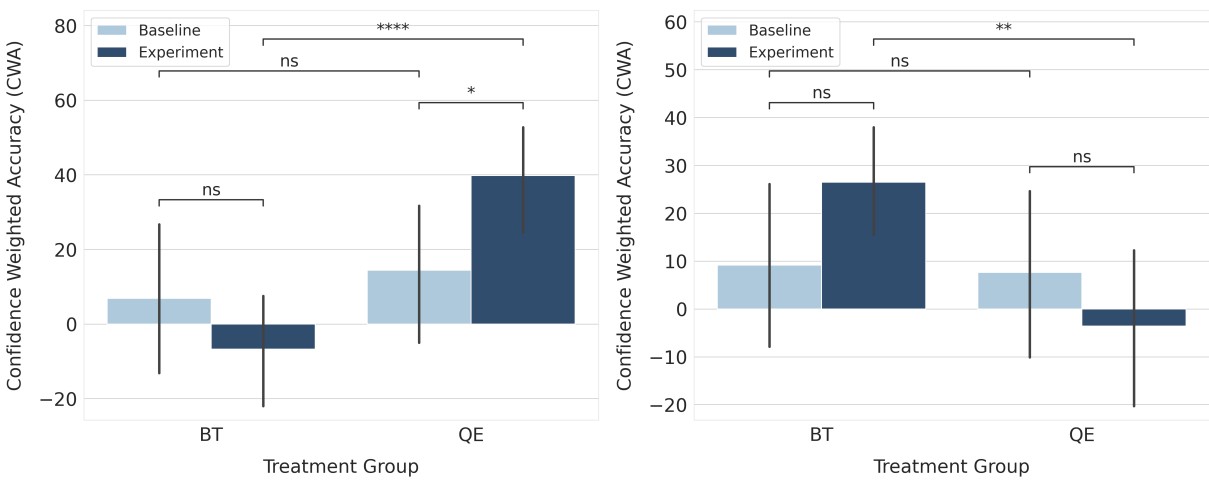

(a) Correct Identification of Adequate Translations      (b) Correct Identification of Inadequate Translations

Figure 4: The QE treatment group had an overall higher CWA in correctly relying on MT systems when (a) translations were adequate whereas the BT treatment group had a higher CWA in judging (b) inadequate translations. ns: not significant;∗∗: significant with p-value of 0.01; ∗ ∗ ∗∗: p-value of 0.0001.

findings that BT-based metrics can complement off-the-shelf supervised QE systems in automatic QE settings (Agrawal et al., 2022). However, it remains to be seen whether presenting the BT output itself is needed, or whether providing finer-grained QE feedback could also play that role.

In the post-survey, we asked physicians how they would want translation quality estimates to be provided. Respondents reported that they would most want a binary indicator of whether the translation was correct, an explanation of where the errors occurred, and error categories relevant for medical purposes. Our study showed the potential of QE using one specific scheme, and motivates future work refining the **presentation of QE feedback** to best support physician needs.

Our study took place in the midst of many discussions around the use of **large language models (LLMS) in clinical settings**, including their

potential to assist physicians in writing discharge instructions (Arora and Arora, 2023), and the development of dedicated clinical language models, such as GatorTron which was trained on over 90 billion words from electronic health records (Yang et al., 2022). Our survey asked physician respondents how they perceived the use of ChatGPT in their workflows (Homolak, 2023; Dave et al., 2023; Li et al., 2023). We found 77% of physicians would use ChatGPT in their clinical workflows, barring regulatory requirements and legal implications. More specifically, 78% responded they would use ChatGPT to summarize patient notes, including symptoms and treatment plans, 60% responded they would use it to answer patient questions, 32% for clinical decision support and to make evidence-based recommendations, and 12% vowed to never use it in their actual clinical workflows. This further highlights the urgent need for NLP work that de-

velops appropriate mechanisms for people to make appropriate use of language generation tools beyond MT in clinical settings.

## 7 Conclusion

We conducted a randomized experiment with 65 English-speaking physicians to test how quality estimation and backtranslation interventions impact their ability to decide when to rely on imperfect MT outputs. We found that the QE group had a significantly higher accuracy in their overall decision to give or not give a translation to a patient, while the BT group detected critical translation errors more effectively. This study paves the way for future work designing methods that combine the strengths of QE and BT, and contributes a human-centered evaluation design that can be used to further improve MT in clinical settings.

More broadly, this work provides support for the usefulness of explanations for helping people rely on AI tools effectively in a real-world task. Our explanations focus on providing actionable feedback rather than explaining the internal workings of the MT model, aligning with recent calls to rethink explainable AI beyond interpretability (Miller, 2023). It remains to be seen how to provide feedback that gives users more agency in appropriately using imperfect MT outputs, and how to design for appropriate reliance on other NLP tools in clinical settings, including large language models.

## Limitations

The study is naturally limited to specific experimental conditions. For example, we evaluated translation for a single language pair, and MT system. While this is motivated by real-world communication needs in hospitals in the United States, it is unclear how these findings would generalize to other language pairs, including translation into, from, or between languages that are underrepresented in the MT training data which would likely lead to lower translation quality, as well as translation between more closely related languages where users might be able to exploit cognates or other cues to assess translation quality even if they only speak one of the two languages involved. Future research will consider the combined use of multiple interventions and larger sample sizes of sentences.

Emergency department discharge instructions represent one of many forms of communication between physicians and patients, and future work needs to explore how MT can be used for other settings. Additionally, our study focused on physicians' reliance on MT, but successful communication naturally requires taking into account the patient's perspective as well, which we will consider in future work.

## Ethics Statement

This work was conducted in line with the EMNLP Ethics Policy. Models, datasets, and evaluation methodologies used are detailed in our Methods section. The discharge instruction sentences went through a rigorous de-identification process to ensure no patient information was compromised. The use of these discharge notes were approved to be released by the university hospital that we obtained them from. Our study was approved by our university's Institutional Review Board (IRB). Physicians gave their consent to participate in the study, and were compensated for their time.

## Acknowledgements

We would like to thank the physician participants for their time and the anonymous reviewers for their constructive feedback. We would also like to thank Coye Cheshire for early feedback on the study design. This work is supported by NSF Fairness in AI Grant 2147292.

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

# A  Survey Design

We provide an example of one sentence from a single discharge note in each of the two interventions: BT and QE. Physicians in both treatment groups received the baseline questions prior to receiving the same sentence accompanied by their quality feedback.

## A.1  Baseline

**Instruction**  Google Translate is a translation system that allows users to automatically translate text from one language to another. Automatic translation systems like Google Translate; however, can sometimes make errors.

**Please respond to the following questions barring a lawsuit and regulatory requirements around showing an imperfect translation to a Chinese-speaking patient. You are solely focused on patient comprehension of discharge instructions as it pertains to clinical outcomes.**

**Please answer each question assuming that you have already reviewed the sentence from the discharge instruction with a patient using a medical interpreter.** Use your best judgment when answering questions, and do not consult outside sources.

You will be shown 28 sentences from a total of six discharge instructions.

**Discharge Note**  The following sentence was passed through Google Translate: *You were in the hospital because you had a loss of consciousness after getting an IV placed.* Google Translate generated the following **Chinese translation** for the sentence above: 您之所以在医院，是因为放置静脉注射后丧失了意识。

**Question 1:   Would you give the Chinese translation to a patient who only reads Chinese?**

**Question 2:**   How confident are you in your decision to give (or not give) the above translation to a patient? Likert scale: Confidence level (1=Not Confident; 5=Very Confident)

## A.2  BT

**Instruction**  In the following section, you will be shown a Chinese translation of an English sentence that was passed through Google Translate. You will then be shown the English translation of that same Chinese sentence after it was passed through Google Translate again.

**Please respond to the following questions barring a lawsuit and regulatory requirements around showing an imperfect translation to a Chinese-speaking patient. You are solely focused on patient comprehension of discharge instructions as it pertains to clinical outcomes.**

You will be shown 28 sentences from a total of six discharge instructions.

**Discharge Note**  The following sentence was passed through Google Translate: **You were in the hospital because you had a loss of consciousness after getting an IV placed.** Google Translate generated the following **Chinese translation** for the sentence above: 您之所以在医院，是因为放置静脉注射后丧失了意识。 Google Translate translated this text back into English as: **You are in the hospital because you lose consciousness after the vein injection.**

**Question 1:   Would you give the Chinese translation to a patient who only reads Chinese?**

**Question 2:**   How confident are you in your decision to give (or not give) the above translation to a patient? [Likert scale: Confidence level (1=Not Confident; 5=Very Confident)]

**Question 3:**   If a monolingual Chinese patient were to read the previous sentence, do you think the patient would understand the discharge instruction sentence? [Likert scale: Comprehension ability (1=Patient would understand none of the meaning; 5=Patient would understand all of the meaning)]

### A.3 QE

**Instruction** A quality estimation system can estimate the quality of the translated text by comparing the machine-generated translation (e.g. Chinese) to the original text (e.g. English). This model is trained to try to estimate how a human would rate the quality of a translation.

The system will generate one of the three ratings:

- The meaning of the translation is consistent with the source.

- The translation retains most of the meaning of the source.

- The translation preserves only some of the meaning of the source but misses significant parts.

**Please respond to the following questions barring a lawsuit and regulatory requirements around showing an imperfect translation to a Chinese-speaking patient. You are solely focused on patient comprehension of discharge instructions as it pertains to clinical outcomes.**
You will be shown 28 sentences from a total of six discharge instructions.

**Discharge Note** The following sentence was passed through Google Translate: **You were in the hospital because you had a loss of consciousness after getting an IV placed.** Google Translate generated the following **Chinese translation** for the sentence above: 您之所以在医院，是因为放置静脉注射后丧失了意识。 A quality estimation automatic translation system estimated the translation quality as: **The meaning of the translation is consistent with the source.**

**Question 1:** **Would you give the Chinese translation to a patient who only reads Chinese?**

**Question 2:** How confident are you in your decision to give (or not give) the above translation to a patient? [Likert scale: Confidence level (1=Not Confident; 5=Very Confident)]

**Question 3:** If a monolingual Chinese patient were to read the previous sentence, do you think the patient would understand the discharge instruction sentence? [Likert scale: Comprehension ability (1=Patient would understand none of the meaning; 5=Patient would understand all of the meaning)]

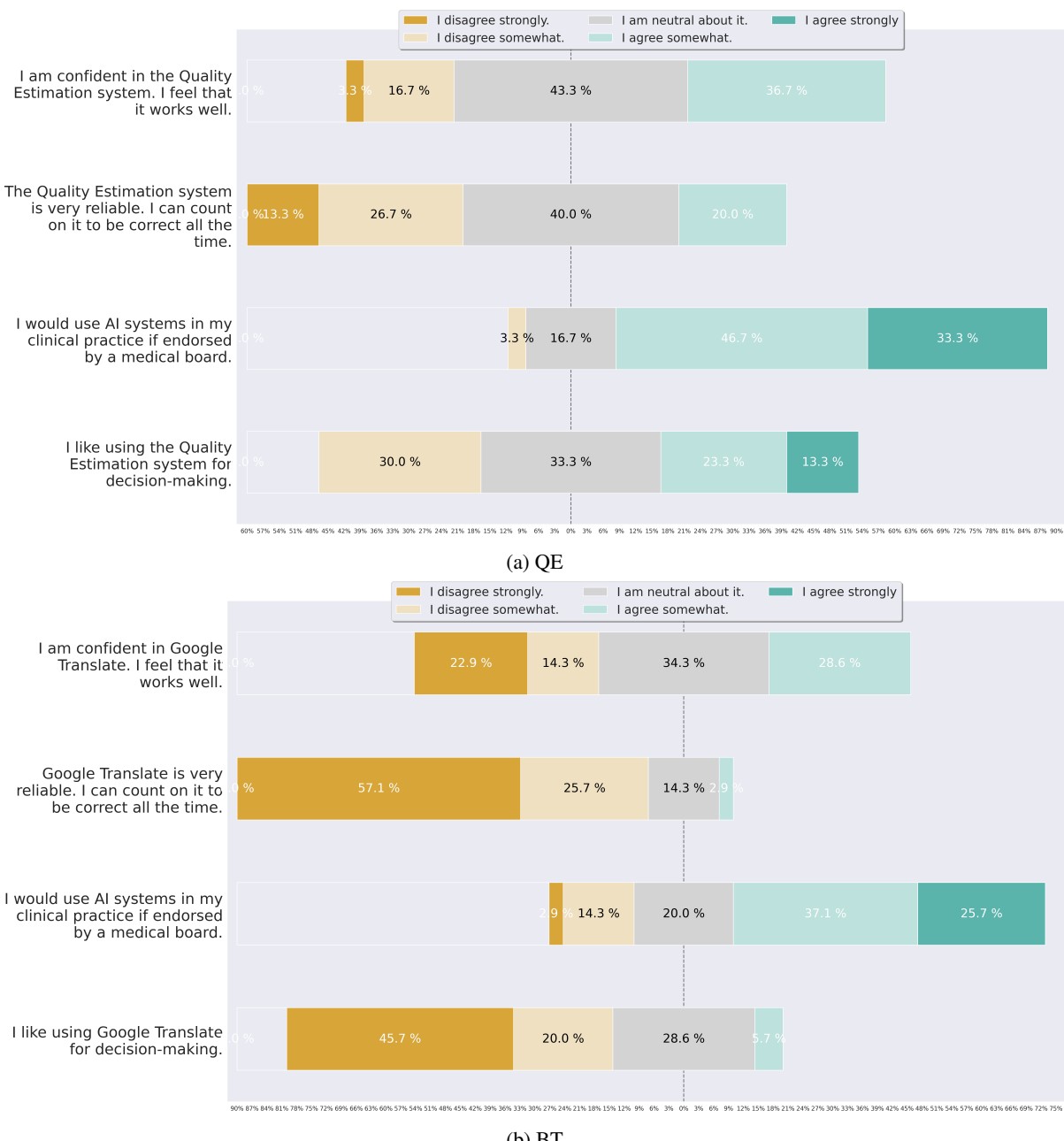

(a) QE

(b) BT

Figure 5: Post-Survey Physicians' Trust Analysis on the use of QE and MT systems in Clinical Workflows.