# OpenReview forum: "Physician Detection of Clinical Harm in Machine Translation: Quality Estimation Aids in Reliance and Backtranslation Identifies Critical Errors"
_EMNLP/2023/Conference — EMNLP 2023 Main_

### Official Review · Reviewer_6psY · 2023-08-04

**Soundness:** 4

**Excitement:**

3: Ambivalent: It has merits (e.g., it reports state-of-the-art results, the idea is nice), but there are key weaknesses (e.g., it describes incremental work), and it can significantly benefit from another round of revision. However, I won't object to accepting it if my co-reviewers champion it.

**Paper Topic And Main Contributions:**

This paper experimentally investigates the impact of QE and BT on the decision-making process of physicians when providing translations for discharge instructions to non-English-speaking patients. The authors provide QE or BT results with discharge instructions to a group of 65 English-speaking physicians and analyze their final decision results.

According to the experimental results, the physician group that referred to QE information reported higher accuracy in making decisions about providing translation results to patients. When utilizing BT, they more effectively detected cases with higher risk. The authors also suggest the possibility of complementary utilization of QE and BT.

**Questions For The Authors:**

1. Why do you think this paper should be addressed at the EMNLP conference? The practicality is acknowledged in terms of focusing on investigating LM's impact on the medical domain. However, the paper only addresses the influence of QE and BT on physicians' decision-making. This is considered more appropriate to be addressed in the medical domain.

2. What is the significance of the paper? What takeaways do the authors want to reveal?

3. Why the authors only deal with English-Chinese pairs? There are many possible language pairs.

**Reasons To Accept:**

This paper is well-motivated. Especially in the medical domain, where human health and even life are directly related, even a single translation error carries significant risk. To prevent safety issues caused by simple translation errors, the practical analysis of the roles of information (QE or BT) for physicians is valuable.

**Reasons To Reject:**

1. The motivation of this paper is well-established; however, there are doubts about whether the content of this paper is suitable for the EMNLP conference. This paper is not about how translation errors of language models appear in the medical field or how to produce correct translations through rewriting, etc. The paper does not explore the technical aspects and focuses only on the practical aspects of the utilization of Commercialized MT and QE systems, lacking in-depth technical details.

2. Lines 515-539 cover the content related to LLM in this paper. Recently, LLM has shown outperforming performance, and the importance of its wide societal utilization has been emphasized by an increasing number of users. It is regrettable that this paper only utilizes the Commercialized MT system (Google Translate), not containing LLM. Although experiments related to ChatGPT are presented in those lines, they are not comprehensive and only rely on simple sociological statistics.

3. The data statistics are not provided in the paper (Section 4.1). Due to the possibility of different evaluation or interpretation depending on the data, I consider it important to provide statistics about the data in experiments. Especially since the authors conducted new annotations, this aspect is further emphasized.

4. This paper only deals with English-Chinese language pairs. The motivation presented by the authors could be applicable to all language pairs, but there is no specific reason given for choosing Chinese, and they did not cover multiple languages. In fact, it is understandable that experimentally dealing with multiple languages could have been challenging. However, there should have been a reason or explanation for utilizing Chinese in the study.

**Reproducibility:**

3: Could reproduce the results with some difficulty. The settings of parameters are underspecified or subjectively determined; the training/evaluation data are not widely available.

**Reviewer Confidence:**

3: Pretty sure, but there's a chance I missed something. Although I have a good feel for this area in general, I did not carefully check the paper's details, e.g., the math, experimental design, or novelty.

---

> ### Author Rebuttal · Authors · 2023-08-29
>
> Thank you for your feedback; it has definitely strengthened our paper. We have addressed your comments below:
>
> 1. We believe that this paper is well-suited for the Human-Centered NLP track included in the EMNLP 2023 Call for Papers.  We view our work as aligning with a growing trend toward NLP research that accounts for the people who interact with, and are impacted by, NLP technology, as can be seen for instance in the papers highlighted in the Human-Centered NLP Special Theme at NAACL 2022, and in the Human-Centered NLP courses by Diyi Yang at Stanford or by Sherry Tongshuan Wu at CMU, which include many ACL/EMNLP publications. The long-term goal of this line of work is to build and continually improve the design of the systems to meet the concerns of users, and move beyond solely performance and accuracy measurements.
>
> - We argue that our findings are relevant to a broad segment of the MT/NLP community, beyond the sub-community working on medical applications. We studied the effectiveness of two distinct NLP techniques for giving lay users feedback on the quality of MT. Characterizing the impact of MT errors rigorously requires situating our study in a specific context of use: otherwise, it is unclear how to reliably decide whether an error is critical or not.  We selected physician’s decision-making as a case study, because it corresponds to a real-world need,  in addition to providing a context where the consequences of MT errors are clear. Our study paves the way for future work evaluating QE mechanisms in a more diverse range of settings.
>
> 2. There are a few key takeaways we want to highlight with this work. We found that the physicians exposed to the QE intervention had a significantly higher Confidence Weighted Accuracy in their overall decision to give or not give a translation to a patient, while the BT group detected critical clinically harmful translation errors more effectively. These findings not only provide insight into how currently imperfect MT systems can be leveraged by physicians in medical settings when interacting with patients with limited English proficiency, but it also serves to motivate further technical work in developing QE and MT systems that are informed by the needs of users based off of our human-centered evaluation.
>
> - More broadly, this research advances our understanding of explainability in NLP and more broadly AI systems. Our work provides support for the usefulness of technical interventions that help people rely on AI tools effectively, in a real-world task. Our interventions focus on providing actionable feedback rather than explaining the internal workings of the MT model, aligning with recent calls to rethink explainable AI beyond interpretability. Future work can investigate how best to provide feedback that gives users more agency in appropriately using imperfect MT outputs (our work provides a foundation for this), and how to design for appropriate reliance on other NLP tools in clinical settings, including large language models. Our study examined commercialized MT systems because such systems are being widely used by physicians in clinical settings. LLMs are gradually being deployed in the medical context, and future work could investigate how these models such as ChatGPT are being used by physicians in practice.
>
> 3. We chose English-Chinese, because Chinese is a high-demand language pair that is often needed in clinical settings in the United States (Khoong et al. 2019), and is a linguistically interesting pair involving translation across distant languages. While the general translation quality is expected to be reasonably strong for this high-resource language pair, the FLORES English-Chinese (simplified) devtest has a translation quality as measured by BLEU as 38.5 (Goyal et al. 2022). This pair is also well-studied in the MT literature, which influenced our motivation to use Chinese.
>
> 4. In reference to comments around reproducibility, we are committed to releasing: a) the Emergency Department (ED) discharge instructions, b) the translated sentences that were shown to the 65 physicians in our experiment, c) the survey responses from physicians on whether to give a translation to a patient or not as well as their confidence in doing so, and d) the bilingual physician gold annotations labeling adequacy and clinical risk. We will release this data upon acceptance of the paper.

---

### Official Review · Reviewer_fR6t · 2023-08-04

**Typos Grammar Style And Presentation Improvements:** Missing text after line 368 and thus …
**Soundness:** 4

**Excitement:**

4: Strong: This paper deepens the understanding of some phenomenon or lowers the barriers to an existing research direction.

**Paper Topic And Main Contributions:**

This paper presents a study on the use of machine translation in US emergency rooms to translate discharge instructions to patients not speaking English. It compares the use of a quality evaluation tool vs. displaying the text back translated into English using the same machine translation tool.
While the test data is limited in size, the statistic evaluation seems to be correct. The study finds that QE is efficient at helping practicians decide to deliver the translation to the patient, while BT helped to detect dangerous errors.

Suggestion: use a different machine translation tool for back translation than the one used for the initial translation. This would allow reducing the risk of cancelling the translation errors.

**Questions For The Authors:**

I suppose that the double-one in (1) is in fact the same as sign(s) in (2)?

**Reasons To Accept:**

This study is interesting because it shows how to improve a real use of a NLP tool in a real world situation. It is well made and presents convincing results.

**Reasons To Reject:**

There is only 6 discharge instructions and 28 sentences for 5 important elements. So, very few sentences for each element. Is it enough to drive definitive conclusions on the use of QE and BT in this setting?

The labelling of translation pairs seem to be missing after line 368. It makes table 1 hard to interpret.

I don't understand why a third experimental situation has not been studied: showing both QE and BT to physicians. This could (or not…) have enhanced both results. I was on the verge of seeing this as a blocking point.


**Reproducibility:**

5: Could easily reproduce the results.

**Reviewer Confidence:**

4: Quite sure. I tried to check the important points carefully. It's unlikely, though conceivable, that I missed something that should affect my ratings.

---

> ### Author Rebuttal · Authors · 2023-08-29
>
> Thank you for your feedback on our paper. We appreciate you taking the time, and have addressed your comments below:
>
> - We chose 28 sentences across 6 discharge instructions to use in our study. Our focus in this paper is on physician perception of BT and QE interventions, and thus we prioritized a larger human subject sample in our experiment to make more robust conclusions about the effects of these techniques on physician users. We ensured; however, through our bilingual physician gold annotation process, that there was a diverse sample of sentences with varying levels of clinical risk in our experiment that represent a range of potential real-world use cases. This contrasts to typical MT evaluation methods where the goal is to assess the quality of an MT system and there are a larger set of text samples presented to a small number of human subjects. Expanding the dataset is an important area for future work that we have added to the paper.
>
> - Thank you for pointing out the labeling of translation pairs missing after line 368. We have fixed the placing and reference to the table in our paper.
>
> - We agree that an interesting intervention to study would be to combine QE and BT. In this paper, we specifically focused on the individual contributions of each technique on appropriate physician reliance on machine translation.  Our decision to focus on these two interventions was influenced by prior work suggesting that BT would not be effective, and our ability to recruit enough practicing physicians. We have emphasized the complementary effects of each intervention as an important future direction of work.
>
> - In (1), the function used is an indicator function that takes on a value of 1 if the physician classified the sentence correctly, and 0 if the physician was incorrect. This results in a classical definition of accuracy (i.e. percentage of correct answers). The downside to this metric is that it only considers correct answers and does not account for confidence. In our definition of Confidence Weighted Accuracy (CWA), (2), we use a sign function, that also indicates the sign for a correct or incorrect response and weighs it by the confidence of that answer. Confidence Weighted Accuracy is a commonly used metric in studies like ours.

---

### Official Review · Reviewer_m7Bm · 2023-08-04

**Soundness:** 5

**Excitement:**

5: Transformative: This paper is likely to change its subfield or computational linguistics broadly. It should be considered for a best paper award. This paper changes the current understanding of some phenomenon, shows a widely held practice to be erroneous in someway, enables a promising direction of research for a (broad or narrow) topic, or creates an exciting new technique.

**Paper Topic And Main Contributions:**

This paper explores the effectiveness of quality estimation (QE) and backtranslation (BT) to support subject matter experts in deciding when (not) to rely on machine translation (MT). Specifically, the authors designed and conducted a between-subjects experiment in which 65 physicians were randomised into two groups and, with the support of either QE or BT, indicated with what confidence they would or would not give a translation to a patient. Results show that the two decision support aids are complementary: the use of QE lead to significantly higher overall accuracy in deciding whether or not to use MT, while the use of BT lead to significantly higher accuracy in detecting critical translation errors.

**Questions For The Authors:**

* You note that physicians were screened for knowledge of English and Chinese in Line 301f. How did the screening work?
* There is one sentence I can't entirely follow in Line 373f: ‘Of the nine translations that are deemed by bilingual experts to cause moderate to life-threatening harm to users, COMET results in six false positives.’ I tried hard but failed to relate this information to the numbers presented in Table 1 – could you explain where exactly the six false positives are in the table?
* What is XAI (Line 379)?

**Reasons To Accept:**

This paper is strongly relevant to the NLP community and the MT community in particular. Automated decision support aids such as QE receive a lot of attention, and QE in particular is mainly evaluated in comparison to human judgements of MT quality for ‘general’ texts. Outside of professional translation, QE (and related) methods clearly have a vast potential in informing decisions about whether or not to use MT in real-life tasks, but there is little to no published evidence that NLP-based decision support is effective in practice. This paper contributes a neatly designed case study in a field where MT is used frequently and where errors can have severe consequences. It also shows that ‘simple’ methods such as BT can complement ‘fancier’ models such as SOTA QE, and that the research community is well-advised to not drop the former off the radar.

**Reasons To Reject:**

TL;DR: This is a very strong paper. I will fight to have it accepted.

Other reviewers might be tempted to suggest that this paper would be better suited for publication at another venue, such as a human–computer interaction (HCI) conference. However, I strongly advise the (S)ACs and PC members to accept it for publication at EMNLP. As rightly pointed out, for example, by Stanford NLP's Diyi Yang in the ACL 2023 panel on The Future of Computational Linguistics in the LLM Age, NLP technology is easier than ever to use by people, so we – the NLP community – should care about how people use, and can be enabled to make the best possible use of, NLP technology. Earlier EMNLP conferences also featured papers that focussed on the perception of MT. For instance, https://aclanthology.org/D18-1512/ garnered significant media attention and continues to be cited regularly.

**Reproducibility:**

3: Could reproduce the results with some difficulty. The settings of parameters are underspecified or subjectively determined; the training/evaluation data are not widely available.

**Reviewer Confidence:**

4: Quite sure. I tried to check the important points carefully. It's unlikely, though conceivable, that I missed something that should affect my ratings.

**Typos Grammar Style And Presentation Improvements:**

First off: congratulations on the crystal clear writing.

Some (mostly minor) suggestions for improvement:
* I'm not sure if pointing out Google Translate's BLEU on FLORES is very helpful since – as with every black box system – it cannot be ruled out that these test data are part of the training data.
* Line 362f: Revise punctuation/bracketing

---

> ### Author Rebuttal · Authors · 2023-08-29
>
> Thank you so much for your time and encouraging feedback. Addressing your comments below:
>
> - In terms of screening, we used Qualtrics to administer our experiment. After providing context on the goal of the study, we included a demographic information section in our survey form where we asked physician participants if English was their first language, and how many other languages they spoke. We stated in bold if they spoke Chinese, they were ineligible to participate in the study. This inclusion criteria for the experiment was also included in our recruitment emails, and we manually filtered out any participants who may have missed this statement and wrote they spoke Chinese in the “other languages” text box.
>
> - In reference to Table 1, we updated our paper to further clarify the COMET-QE labels of 6/9. We added this text: “Table 1 shows the breakdown of QE labels for clinical risk and adequacy assessment on the gold annotation data. COMET-QE achieves an accuracy of 73% and 66% on detecting adequate and clinically insignificant translations, respectively. However, COMET-QE was not effective at detecting incorrect translations that caused clinical risk. Of the nine translations that are deemed by bilingual experts to cause moderate to life-threatening harm to users (columns “moderate”, “high”, and "life-threatening" in Table 1), COMET-QE rated 6/9 as having “consistent” or “most preserved” translations. The QE labels provided are thus accurate enough to be potentially useful, yet realistically imperfect, as expected of explainable artificial intelligence (XAI) interventions.”
>
> - We have clarified in the paper that XAI refers to explainable artificial intelligence.
>
> - In reference to “pointing out Google Translate's BLEU on FLORES…” we included this number to provide a rough indicator of the translation quality of Google Translate.

---

### Meta-Review · Area_Chair_R3PN · 2023-09-18

**Recommendation:** 5

**Metareview:**

This paper studies how physicians might or might not trust the outputs of machine translators to provide these to their patients. A human study was carried out to see how two different tools, quality estimation and back translation, compare in helping physician decide if they will use the MT outputs.

As one of the reviewers has pointed out, this work makes an important contribution in showing how tools such as QE can be used in real applications in more technical fields where there might be severe consequences for any translation errors. All reviewers agreed that this paper has strong soundness, and some reviewers found it strongly exciting.  Still, some suggestions have been made on how to further improve the writing of the paper.

Some of the limitations of this work include the fact that only 6 discharge instructions with 28 sentences were used. Also, the fact that only the English to Chinese translation pair was considered in the study. In a way, the authors need to consider these to these two factors before making any strong claims on the generality of their study. Also, there seems to be a missing opportunity to study the usage of QE and BT together. Adding this condition would have made the study even stronger.

Overall, this paper is good, and I am recommending acceptance to the Main conference.

---

### Decision · Program_Chairs · 2023-10-07

**Decision:**

Accept-Main

**Comment:**

This paper studies how physicians might or might not trust the outputs of machine translators to provide these to their patients. A human study was carried out to see how two different tools, quality estimation and back translation, compare in helping physician decide if they will use the MT outputs.

As one of the reviewers has pointed out, this work makes an important contribution in showing how tools such as QE can be used in real applications in more technical fields where there might be severe consequences for any translation errors. All reviewers agreed that this paper has strong soundness, and some reviewers found it strongly exciting.  Still, some suggestions have been made on how to further improve the writing of the paper.

Some of the limitations of this work include the fact that only 6 discharge instructions with 28 sentences were used. Also, the fact that only the English to Chinese translation pair was considered in the study. In a way, the authors need to consider these to these two factors before making any strong claims on the generality of their study. Also, there seems to be a missing opportunity to study the usage of QE and BT together. Adding this condition would have made the study even stronger.

Overall, this paper is good, and I am recommending acceptance to the Main conference.